# Geographical characteristics and formation mechanisms of smallpox epidemics in Hubei Province, China, 1488–1949

**Yuxin Zeng, Zhiyu Chen, Xihao Yan, Shengsheng Gong, Tao Zhang** *

Hubei Key Laboratory for Geographical Process Analysis and Simulation, Central China Normal University, Wuhan, Hubei, China

* 419448202@qq.com

## Abstract

Smallpox is a highly contagious and ancient disease influenced by natural and social factors. These factors led to the wide spread of smallpox in Hubei Province of China during the historical period. We conducted the spatial and temporal distribution patterns of smallpox epidemics and their formation mechanism in Hubei Province of China during 1488–1949. Based on epidemic history and environmental data, we used M-K test, wavelet analysis, spatial autocorrelation model, epidemic center of gravity model and geographically weighted regression models. In terms of temporal changes, the earliest smallpox in Hubei Province can be traced to the Ming Dynasty. Smallpox epidemics in the Republic of China showed fluctuating and changing trends; in 1939, incidences of smallpox grew abruptly. Smallpox epidemics in the Republic of China occurred on a fluctuating cycle of two time scales: 8 years and 18 years. The epidemic season was mainly spring and summer. Smallpox epidemics in Hubei Province had a wide spatial scope and exhibited spreading and diffusion characteristics; three towns of Wuhan, Suixian and Yichang were the centers of the epidemics. Smallpox epidemics exhibited significant spatial concentrations; high concentration areas occurred mainly in Wuchang, Hankou and Hanyang. The center of gravity of the epidemics exhibited a small swing from east to west and gradually shifted to the west. River networks, road networks, wars and other factors promoted smallpox epidemics; river networks and war factors were significant in eastern Hubei Province; road network factors were significant in southern Hubei Province; and droughts somewhat inhibited smallpox epidemics in western Hubei Province. Temperature fluctuations, droughts and floods, and war outbreaks played dominant roles in the temporal characteristics of smallpox epidemics in Hubei Province, and topography, population distribution and population movement played dominant roles in the spatial distribution pattern of smallpox epidemics in Hubei Province. We must establish and improve an epidemic monitoring and early warning system, pay attention to key areas, strengthen inspection and quarantine, stockpile smallpox vaccines,

**Data availability statement:** All relevant data are within the paper and its Supporting Information files

**Funding:** This work was supported by the National Natural Science Foundation of China (42371265, 41801141, 42471285),and the National Social Science Foundation (21VJXT015).The funders have role in study design, data collection and analysis, decision to publish, or preparation of the manuscript.

**Competing interests:** The authors have declared that no competing interests exist.

develop therapeutic drugs, and strengthen prevention of bioterrorism. Our study revealed how smallpox spreads in terms of both spatial and temporal patterns and mechanisms, and based on this, we can propose preventive and control measures against smallpox reemergence and its similar viruses.

## 1. Introduction

Smallpox is a highly contagious and ancient disease caused by the smallpox virus and belongs to China's category B statutory infectious diseases [1]. Smallpox virus belongs to the Orthopoxvirus genus under the Poxviridae family, which is susceptible to humans, mainly through airborne transmission, with high infectiousness and mortality [2]. Smallpox is known to have existed in ancient China, India, and Egypt more than 3000 years ago [3]. In ancient China, smallpox was also called the "pox sore" or the "lu sore," and the earliest record of smallpox symptoms can be found in Ge Hong's "A Handbook of Prescriptions for Emergencies," which suggests that smallpox was introduced to China during the Jianwu period of the Eastern Han Dynasty [4]. Research on smallpox epidemics has focused mainly on the following four aspects. The first aspect is a study on the temporal and spatial distribution of smallpox epidemics. Smallpox initially became common since the Eastern Jin Dynasty [5]. Smallpox was more thoroughly documented in the Sui and Tang dynasties, as it was one of several epidemics that caused pandemics during this period [6]. After the Song and Yuan Dynasties, smallpox became more prevalent in China [7]. After the Ming Dynasty, its range expanded [8]. During the Republic of China, due to continuous wars and frequent disasters, the scale of smallpox epidemics was enormous [9]. Smallpox was still rampant after the founding of the People's Republic of China, but with the popularization of the policy of compulsory pox inoculation, the proliferation and spread of smallpox were effectively controlled [10]. The second aspect is an analysis of the factors influencing smallpox epidemics. Climatic factors strongly influence smallpox epidemics [11]. Under normal circumstances, smallpox viruses are more stable in cooler environments with moderate humidity, while high temperatures prevent smallpox viruses from surviving [12]. Drought and flood disasters often indirectly induce smallpox in the form of a disaster chain. [13]. The prevalence of smallpox is also associated with trade, war, medical knowledge, and immunization [14]. The third aspect is related to the response and impact of smallpox epidemics. All sectors of society had to take corresponding countermeasures and measures to address the spread of smallpox strains. Vaccination via variola to cowpox laid an important foundation for the eradication of smallpox [15]. Smallpox, as a long-standing and prevalent epidemic, has also impacted China's political direction and ethnic relations [16]. The fourth aspect is a study on the history of smallpox in Hubei Province. More than ten main types of urban diseases occurred in Hubei Province during the Republic of China, of which infectious diseases such as smallpox had the greatest impact, as they caused more deaths [17]. After the founding of the People's Republic of China, smallpox was still rampant, and smallpox was prevalent in Yichang Yidu and

Suixian, resulting in many deaths [18]. Changes in temperature had a more significant influence on smallpox epidemics in Hubei Province,which led to a decrease in human immunity and susceptibility to smallpox infection [19]. In summary, the existing studies mainly have the following deficiencies: First, most of them are qualitative studies on the spatiotemporal distribution, influencing factors, and response impacts of smallpox epidemics from a historical perspective, lacking comprehensive quantitative studies from the perspectives of geography, history, and epidemiology. In particular, the analysis of the influencing factors of smallpox is not systematic enough and lacks exploration of the formation mechanism; second, there is still a lack of specialized research on historical smallpox in Hubei, and existing research mainly focuses on cholera [20] and malaria [21]. What are the spatiotemporal characteristics of the epidemic of smallpox in Hubei history? What are the influencing factors and formation mechanisms of its spatiotemporal characteristics? These are the questions we are interested in. Based on historical smallpox data and environmental data, this study was carried out from the perspective of historical medical geography to investigate the spatial and temporal distribution patterns of smallpox epidemics in Hubei Province during the period of 1488–1949 and the underlying mechanism to reveal the relationship between epidemics of infectious diseases and the geographic environment and to provide a scientific basis for the prevention and control of infectious diseases.

## 2. Materials and methods

### 2.1 Study area

Hubei Province is located in the central region of China and is surrounded by mountains from east, west and north; it is low and flat in the middle, with a high proportion of undulating terrain and mountainous hills. In addition to high mountainous areas, most of the region has a subtropical monsoon humid climate, is cold in winter and hot in summer, with variable temperatures in spring and autumn. The Yangtze River and the Han River converge in Hubei Province, which has a dense network of rivers and numerous lakes. Located in the core of central China, Hubei Province is in a very favorable geographical position, with well-developed land and water transport, and is known as the "throughway of nine provinces." Smallpox, cholera, malaria, and other epidemics have occurred frequently in Hubei Province throughout history. Based on the period of the study, this paper used the administrative division of 1946 as the standard mapping year, during which Hubei Province was divided into 71 counties and cities (Fig 1).

### 2.2 Data collection

2.2.1 Smallpox epidemics data. Smallpox epidemics data are collected mainly from the "Compilation of Historical Data on Epidemics in China for Three Thousand Years" [22]. This compilation is rich in historical materials, covering official history, archives, records, anthologies, local chronicles, and modern newspapers and periodicals. In addition, the data were supplemented from the "Wuhan Daily" and "Dagang Daily (Hankou)" during the Republic of China, and a total of 175 historical materials on smallpox epidemics were collected from 1488 to 1949. On the basis of historical data, with counties as the basic unit, we counted the number of counties where smallpox occurred each year and the number of years when smallpox occurred in each county and compiled a chronology of smallpox epidemics in Hubei Province.

2.2.2 Influencing factors data. The natural factors included average annual temperature, annual precipitation, elevation, river network density, droughts and floods. The climate data were obtained from the National Earth System Science Data Centre (http://loess.geodata.cn/), the average annual temperature and average annual precipitation of each county were obtained by extracting and calculating the monthly raster data with a resolution of 1 km from 1900 to 1949. The elevation and river data were extracted from the DEM maps by using ArcGIS 10.7 software. The meteorological disaster data used were obtained from the "Chinese Meteorological Disasters Hubei Volume" [23] and "Climate Historical Data of Hubei Province in the Past Five Hundred Years" [24], the number of years of droughts and floods in each county between 1488 and 1949 was obtained by statistics. Social factors included population density, road network density,

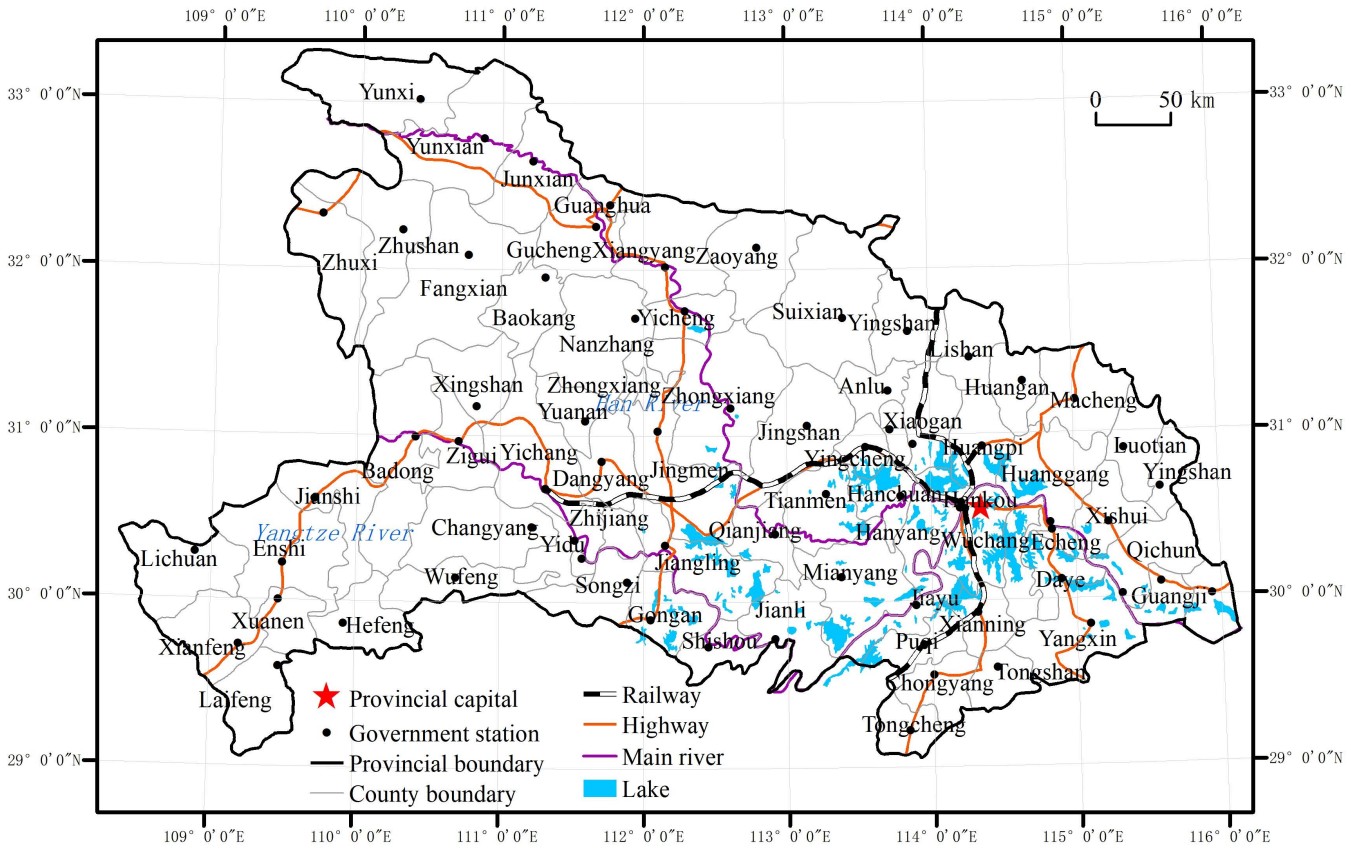

**Fig 1. Administrative division map of Hubei Province in 1946.** Note: The basemap came from United States Geological Survey (https://apps.nationalmap.gov/services/), the map boundary has not been changed. Cartographic software: ArcGIS.

war and turmoil. Population data were based on average population data from the Republic of China and were collected from the "Hubei General Chronicles" [25] "Ministry of Internal Affairs Statistics of Hubei Population in the Five Years of the Republic of China" [26] "Statistical Report on Household Registration Survey of Provinces and Municipalities in the Seventeenth Year of the Republic of China" [27] "Hubei Demography" [28] and "Implementation Minutes of the General Review of Household Registration in the Winter of the Thirty-Five Years of Hubei Population" [29]. The road network density data come from 1935. The road network of the "Detailed Map of Hubei" was vectorized and calculated, and the war data were compiled based on the "Chronology of Chinese War in Past Dynasties" [30] and "The History of Chinese War Volume 4·Republic of China " [31], the number of years of wars in each county between 1912 and 1949 was obtained by statistics.

The smallpox epidemics data and influencing factors data were unified and integrated through data standardization, with years as the basic time unit and counties as the basic spatial unit.

### 2.3 Research methods

**2.3.1 Trend cycle analysis methodology.** The Mann–Kendall (M-K) test was used to analyze smallpox temporal characteristics, and the wavelet analysis method was used to extract the smallpox change cycle. The M-K statistical test for trend is a nonparametric statistical test that can be used to analyze monotonically trending time-series data to determine the change rule of the trend of the time series; this test is not related to the type of sample values and sample distributions, and

the overall results of the analysis are not affected by the interference of a small number of outliers. The method is independent of the sample value and sample distribution type, and the overall analysis results are not disturbed by a few outliers, which is applicable to a variety of nonnormally distributed sample data [32]. Its calculation method and formula are as follows:

For a time series $x_i$ ($i$ =1,2,…,n) with n sample sizes, construct an order column $S_k$, $S_k$ denotes the cumulative number $x_i > x_j$ ($1 \le j \le i$) in the *ith* sample, and define $S_k$ as:

$$S_k = \sum_{i=1}^{k} r_i (k = 1, 2, ..., n), r_i = \begin{cases} 1 & x_i > x_j \\ 0 & x_i \le x_j \end{cases} (j =, 2, ..., i),$$

(1)

Assuming that the time series are independently randomized, the mean and variance of $S_k$ are $E(S_k)=k(k-1)/4$, $Var(S_k)= k(k-1)(2k+5)/72$, respectively, and the formula for the statistic $UF_k$ is defined as:

$$UF_k = \frac{S_k - E(S_k)}{\sqrt{Var(S_k)}} (k = 1, 2, 3, ...n),$$

(2)

Where: $UF_k$ is the standard normal distribution, which is a sequence of statistics calculated by the time series $x$ in order $x_1$, $x_2$,…, $x_n$. Construct the inverse sequence $UB$ by the time series $x$ in reverse order $x_n$, $x_{n-1}$,…, $x_1$, and then repeat the above process. If the value of $UF$ is greater than 0, it indicates an upward trend in the sequence, and less than 0 indicates a downward trend. When they exceed the critical confidence level straight line (the confidence level line is ±1.96 at the test confidence level α = 0.05), it indicates that the upward or downward trend is significant, and the range above the critical line is determined as the time region where the mutation occurs. If there is an intersection of the $UF$ and $UB$ curves and the intersection point is between the critical lines, then the moment corresponding to the intersection point is the time of the onset of the mutation. In this study, Matlab 2020a was used to program and plot the statistics UB and UF of the time series of the number of smallpox counties in Hubei Province to test the trends and mutation time points of smallpox epidemics in Hubei Province.

The wavelet analysis method is a transform analysis method that reveals local temporal features and can respond to changes in data at a specific spatiotemporal location [33]. A time series can be used to display multiple cycles of variation hidden in the time series, thus providing a better picture of the variation in the series over cycles [34]. The basic idea of wavelet analysis is to use a cluster of wavelet functions to represent or approximate a signal or function. Wavelet function refers to a type of function that is oscillatory and can decay quickly to zero, and is the key to wavelet analysis. This study uses the Morlet wavelet function (cmor), which has good locality in the time and frequency domains. It can not only show the fine structure of the time series, but also reveal the periodic changes implicit in the sequence that change over time. This study uses the Matlab 2020a "Complex Contious Wavelet 1-D" tool to calculate the real part of the wavelet coefficients and the wavelet variance of the time series of the number of smallpox counties, and uses the Surfer software to draw contour maps of the real part of the wavelet coefficients.

**2.3.2 Spatial autocorrelation model.** Spatial autocorrelation analysis is an assessment of the correlation of variables with reference to their spatial location, that is, the correlation of a variable with itself in space, and can be used to quantify the extent to which similar features cluster and where this clustering occurs.

The global Moran's I is a measure of the overall clustering of spatial data and is defined as follows:

$$I = \frac{N}{\sum_{i=1}^{N}(x_i - \bar{x})^2} \frac{\sum_{i=1}^{N}\sum_{j=1}^{N} w_{ij}(x_i - \bar{x})(x_j - \bar{x})}{\sum_{i=1}^{N}\sum_{j=1}^{N} w_{ij}},$$

(3)

In the formula, $N$ represents the number of spatial units, that is, the 71 counties and cities in Hubei Province; $x_i$ and $x_j$ represent the smallpox incidence rates in $i$ and $j$, respectively; $\bar{x}$ represents the average smallpox incidence rate in all counties and cities; and $w_{ij}$ represents the spatial weight between $i$ and $j$.

Local spatial autocorrelation statistics are measures of spatial correlation for specific observations. By detecting local spatial clustering around a single observation, "hot spots," regions with high levels of variability, or regions that are not significant according to a single global correlation test can be identified. The formula is as follows:

$$I_{LISA} = \frac{N(x_i - \overline{x})}{\sum_{i=1}^{N}(x_i - \overline{x})^2} \sum_{j=1}^{N} w_{ij}(x_j - \overline{x}),$$

(4)

In this study, the global Moran's I and local Moran's I are used to assess spatial autocorrelation. Moran's I index mainly reflects the magnitude of the correlation of a certain attribute value of the whole dataset in space; its value usually ranges from -1 to +1, with $I<0$ indicating a negative correlation, $I>0$ indicating a positive correlation, and $I$ tends to be close to 0 indicating an irrelevant correlation; the specific expressions of the LISA index are of four types, namely, HL (high-low), LH (low-high), HH (high-high), and LL (low-low). This study used the ArcGIS 10.7 "Spatial Autocorrelation" tool to calculate Moran's I for global spatial autocorrelation analysis, and used the "Cluster and Outlier Analysis" tool to analyze local spatial clustering patterns.

**2.3.3 Epidemic center of gravity model.** The center of gravity is derived from physics and refers to the point at which the force of gravity is uniformly applied to each part of an object. The center of gravity of an epidemic is the point in the spatial plane where the moments of the epidemic in a study area reach equilibrium, and the relocation of the coordinates of the center of gravity of an epidemic can reveal the spatial development trend of the research object [35]. The center of gravity distribution can indicate the degree of equilibrium in the spatial distribution of the number of smallpox-endemic counties, while the center of gravity migration reflects the evolutionary trend of smallpox-endemic geography within a region. Its calculation formula is:

$$\overline{x} = \frac{\sum_{i=1}^{n} x_i z_i}{\sum_{i=1}^{n} z_i}, \quad \overline{y} = \frac{\sum_{i=1}^{n} y_i z_i}{\sum_{i=1}^{n} z_i},$$

(5)

In the formula, $\overline{x}, \overline{y}$ represent the coordinates of the corresponding center of gravity of the smallpox epidemics; the coordinates of the center of gravity of each research unit are $x_i$ and $y_i$; and $z_i$ is the number of years of smallpox prevalence in the study unit. This study used the ArcGIS 10.7 "Mean Center" tool, using the number of smallpox years as weights, to calculate the center of gravity of the smallpox epidemics in Hubei Province during different periods.

**2.3.4 Geographically weighted regression model.** Geographically weighted regression (GWR) is a spatial analysis technique and an extension of the ordinary least squares model to the spatial domain. GWR takes nonstationary variables (e.g., demographic factors and natural environmental characteristics) into account and models the local relationship between these predictors and the outcomes of interest by assessing the deviation of the locally weighted regression coefficients from the global coefficients to enable spatial estimation of regression parameters. Its advantage is that it addresses spatial heterogeneity and has high utility in epidemiology. Its formula is:

$$y_i = \beta_0(u_i, v_i) + \sum_{j=1}^{k} \beta_j(u_i, v_i)x_{ij} + \varepsilon_i$$

(6)

In the formula, $y_i$ is the dependent variable for sample $i$, $(u_i, v_i)$ is the spatial coordinate of the $i$th sample point, $\beta_0$ is the intercept term at $(u_i, v_i)$, $x_{ij}$ is the value of the independent variable $k$ of the $i$th sample, $\beta_j$ is the regression coefficient of $x_{ij}$, and $\varepsilon_i$ is the random error of the $i$th sample point. This study used ArcGIS10.7 "Geographically Weighted Regression" tool to quantitatively analyze the influencing factors of smallpox epidemic in Hubei Province.

## 3. Results

### 3.1. Temporal changes in smallpox epidemics in Hubei Province

**3.1.1 Epidemiological profile.** The earliest smallpox in Hubei Province may date back to the first year of Hongzhi in the Ming Dynasty (AD1488). The "Xiangyang Prefecture Chronicles" from the Wanli period records the following: "In the spring of the first year of Emperor Xiaozong's reign, evil spirits appeared in (Xiangyang), black like mist, like human figures, touching people, causing death among children, and man-made market strikes," [36]. The epidemic is believed to be smallpox and the only suspected epidemic of smallpox during the Ming Dynasty in Hubei Province. Smallpox occurred for 9 years in Hubei Province during the Qing Dynasty, with a frequency of 3.4%. For example, in the 49th year of Kangxi (1710), Fangxian experienced "a severe outbreak of acne and more than a thousand deaths," [22]. In the tenth year of the Guangxu reign (1884), in Jiangxia, "in the sixth month of summer, smallpox was prevalent in the townships of Wuhan," [22]. During the 38 years of the Republic of China, 36 years had smallpox epidemics, with a frequency of 94.74%, of which smallpox was the most serious in 1946, with epidemics occurring in 18 counties. Overall, during the 462 years from 1488 to 1949, 179 years had smallpox epidemics in Hubei Province (Fig 2), for a frequency of 38.7%. Smallpox was scarce during the Ming and Qing dynasties, and smallpox epidemics were more severe during the Republican period.

**3.1.2 Trend change.** Smallpox trends in Hubei were analyzed using the M-K test. Due to the low frequency of smallpox before the Republic of China, only the time series of the number of smallpox counties in each year of the Republic of China was tested (Fig 3). The results showed that the UF value was less than 0 during the period from 1912–1931, indicating that smallpox in the Republic of China had a decreasing trend during the period from 1912–1931, and the UF value exceeded the 0.05 significance threshold during the period from 1920–1924, which indicated that the decreasing trend in this period was very significant; during the period from 1931–1944, it was very significant. The UF and UB curves intersected three times before and after 1939, and the intersection point fell between the critical lines, indicating that the smallpox epidemics began to mutate and grow in 1939.

**3.1.3 Cycle change.** The contour plots of the real parts of the wavelet coefficients can reflect the cyclic variations in the smallpox year series at different time scales and their distributions in the time domain. The smallpox epidemics in Hubei Province between 1912 and 1949 in the Republic of China exhibited two types of cyclic variations at time scales of 5~10 years and 13~28 years (Fig 4). Among them, there were six quasi-oscillatory cycles of light-heavy alternation on the 5~10-year time scale; these cycles behaved stably throughout the time domain of the study and were domain-wide in nature. There were three quasi-oscillatory cycles with light-heavy alternations on the 13- to 28-year time scale, which were also global in nature.

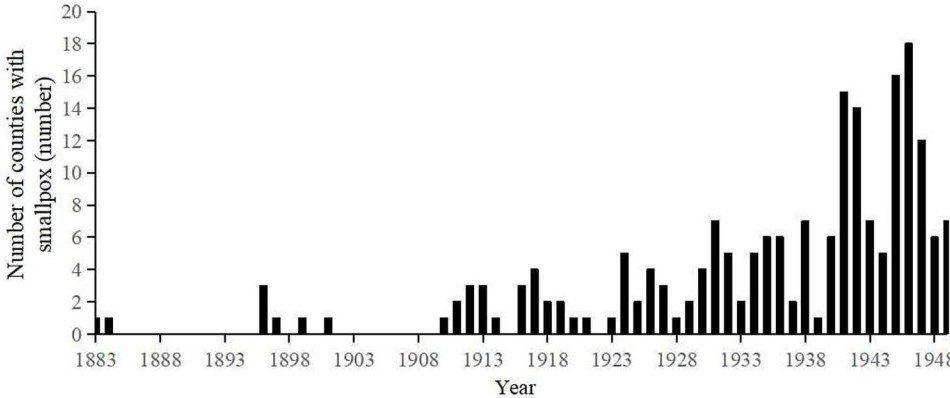

**Fig 2. Annual changes in smallpox counties in Hubei Province from 1883 to 1949.**

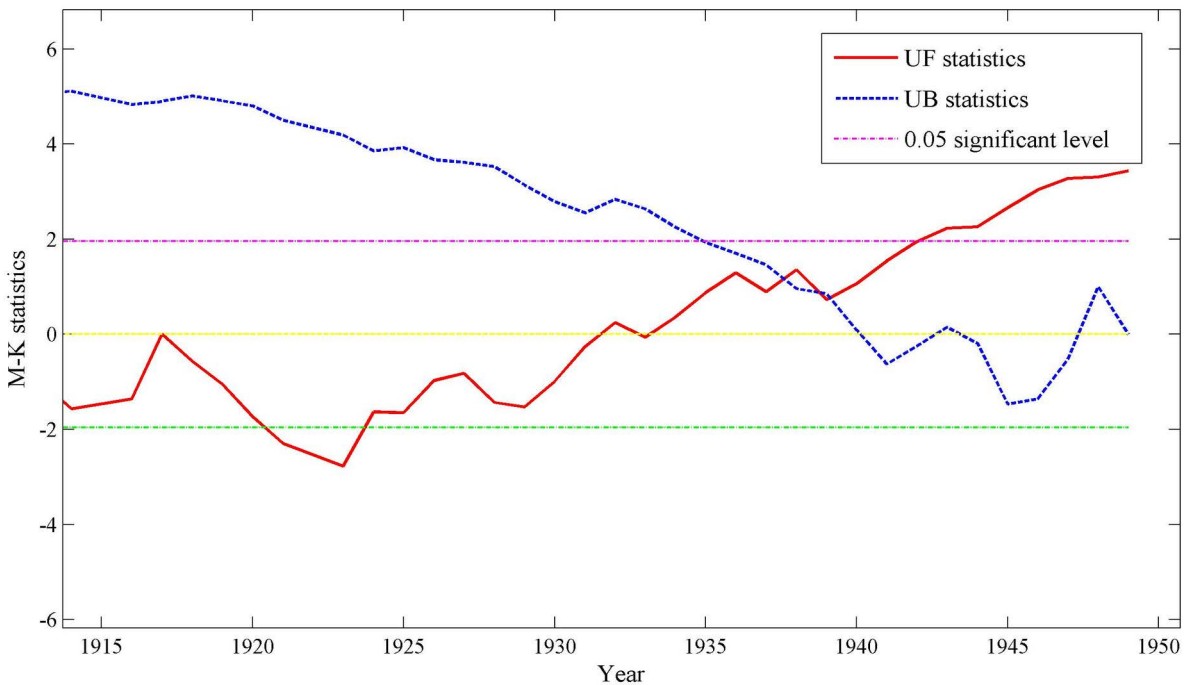

**Fig 3. Mann–Kendall test of counties with smallpox in Hubei Province from 1912 to 1949.**

The wavelet variance plot can reflect the distribution of the fluctuation energy of the smallpox county count sequence over time scales so that the main cycle in the process of smallpox county count change can be determined. Plotting the wavelet variance diagram of the smallpox sequence in Hubei Province shows that there are 2 distinct peaks in the pre-1949 smallpox sequence in Hubei Province, corresponding to the time scales of 8 and 18 years. Among them, the largest peak corresponds to the 8-year time scale, indicating that the strongest cyclic oscillation is approximately 8 years, which is the first main cycle of smallpox change in Hubei Province; the 18-year time scale corresponds to the second peak, which is the second main cycle. The cyclic fluctuations of these two scales control the characteristics of smallpox changes in the whole time domain.

Based on the results of the wavelet variance test, the process lines of the real parts of the wavelet coefficients of the main cycles of the smallpox County number series in Hubei Province on the 8-year scale and the 18-year scale were plotted. An analysis of the process lines shows the characteristics of the average cycle and light-heavy change that existed during smallpox epidemics in Hubei Province. On the 8-year time scale, the average change cycle of the number of smallpox counties was approximately 6 years; the counties experienced 6 cycles of light-heavy conversion, with periods of light epidemics in 1914–1916, 1920–1924, 1928–1929, 1933–1935, 1938–1940, and 1943–1945 and periods of heavy epidemics in 1917–1919, 1925–1927, 1930–1932, 1935–1937, 1941–1942, and 1946–1948. The average cycle of change in the number of smallpox counties on an 18-year time scale was approximately 12 years, with three cycles of light-heavy transitions.

**3.1.4 Seasonal change.** Between 1488 and 1949, 27 years had clear smallpox epidemiological seasons in Hubei Province and 41 cumulative smallpox epidemiological seasons, of which 20 (48.78%) were in spring, 13 (30.71%) in summer, 3 (7.31%) in autumn, and 5 (12.19%) in winter, with spring and summer seasons accounting for 79.49% of the total. The cumulative number of smallpox years with well-defined seasons affected 82 counties—55 counties (67.07%) in spring, 25 counties (30.48%) in summer, 6 counties (7.31%) in autumn, and 7 counties (8.53%) in winter—with more than

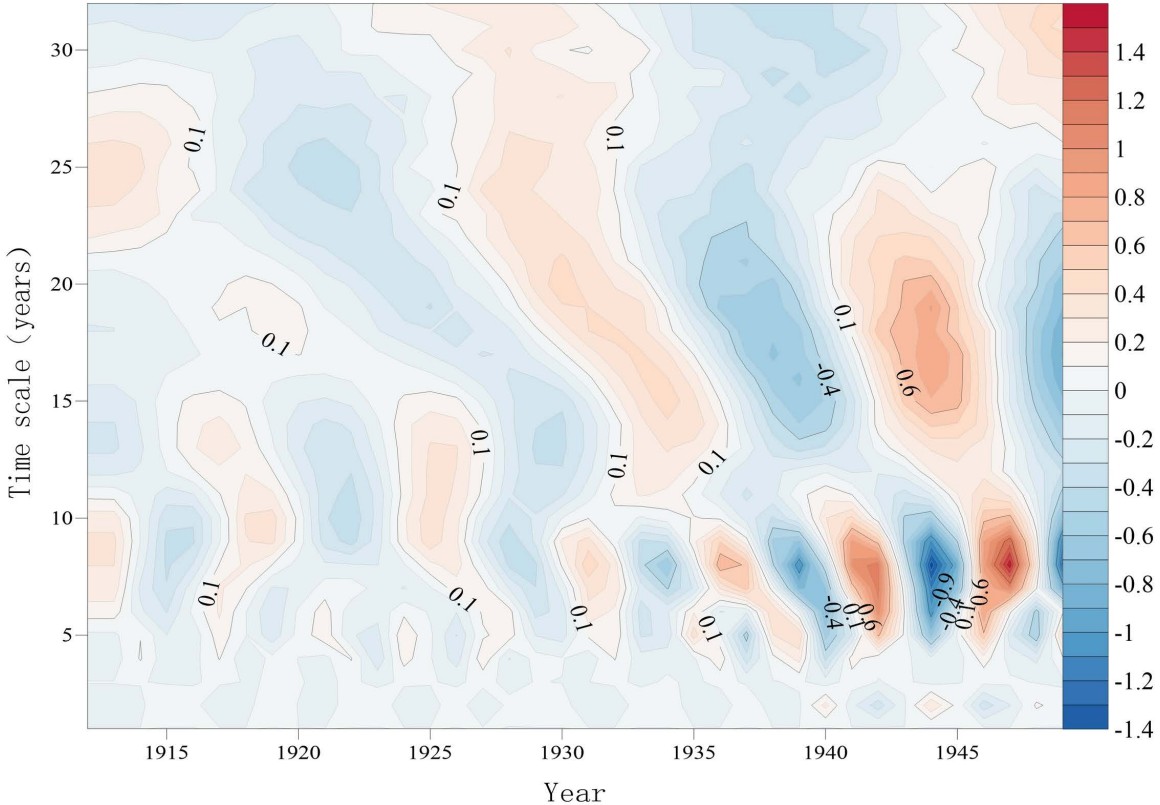

**Fig 4. Contour map of the real parts of the wavelet transform coefficients of the smallpox county series.**

half of the cases occurring in spring and 84.42% of the cases occurring in spring and summer. According to the statistics of the smallpox epidemic seasons and the number of counties with smallpox epidemics in each season, smallpox epidemics in Hubei Province were predominant in the spring and summer seasons, followed by the winter season, and the least common in the autumn season.

### 3.2 Spatial distribution of smallpox epidemics in Hubei Province

**3.2.1 Overall distribution.** During the period of 1488–1909, smallpox was relatively rare (Fig 5a), the spatial distribution was scattered, the epidemic was more severe in the three towns of Wuhan. From 1910–1919, the distribution of smallpox expanded, and the epidemic spread to the northeast of Hubei and the north of Hubei (Fig 5b). From 1920–1929, the distribution of smallpox continued to expand (Fig 5c), and the epidemic spread to the west of Hubei. From 1930–1939, the epidemic began to progress; Wuhan Three Towns and Suizhou County were the hardest hit areas (Fig 5d). From 1940–1949, the epidemic spread to most parts of the province; Wuhan Three Towns in eastern Hubei Province, Suizhou County in northern Hubei Province, and Baokang and Yuan'an in western Hubei Province were the worst-hit areas (Fig 5e). Two smallpox epidemic zones were formed with Wuhan Three Towns and Baokang-Yuanan as the centers, and spread to their peripheries, connected by Suixian. Overall, during the period 1488–1949 (Fig 5f), 64 counties in Hubei Province were endemic to smallpox, with a wide range of epidemiological space; additionally, these counties exhibited the characteristics of spreading and diffusion; the three towns of Wuhan, Suixian and Yichang were the centers of the epidemic in the eastern, northern and western regions of Hubei, respectively.

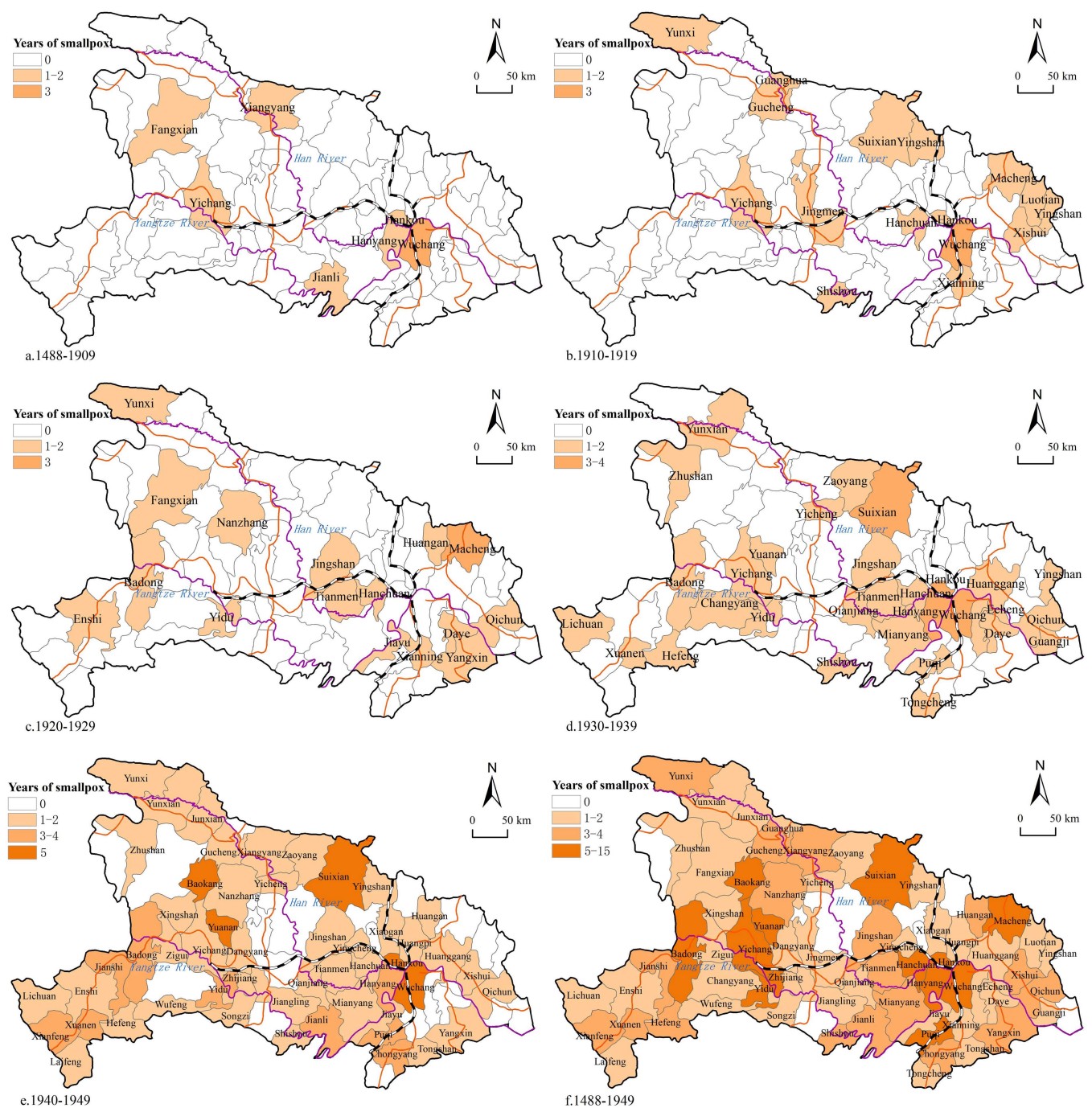

**Fig 5. Overall distribution map of the smallpox epidemics in Hubei Province.** Note: (a)1488-1909.(b)1910-1919.(c)1920-1929.(d)1930-1939. (e)1940-1949.(f)1488-1949. The basemap came from United States Geological Survey (https://apps.nationalmap.gov/services/), the map boundary has not been changed. Cartographic software: ArcGIS.

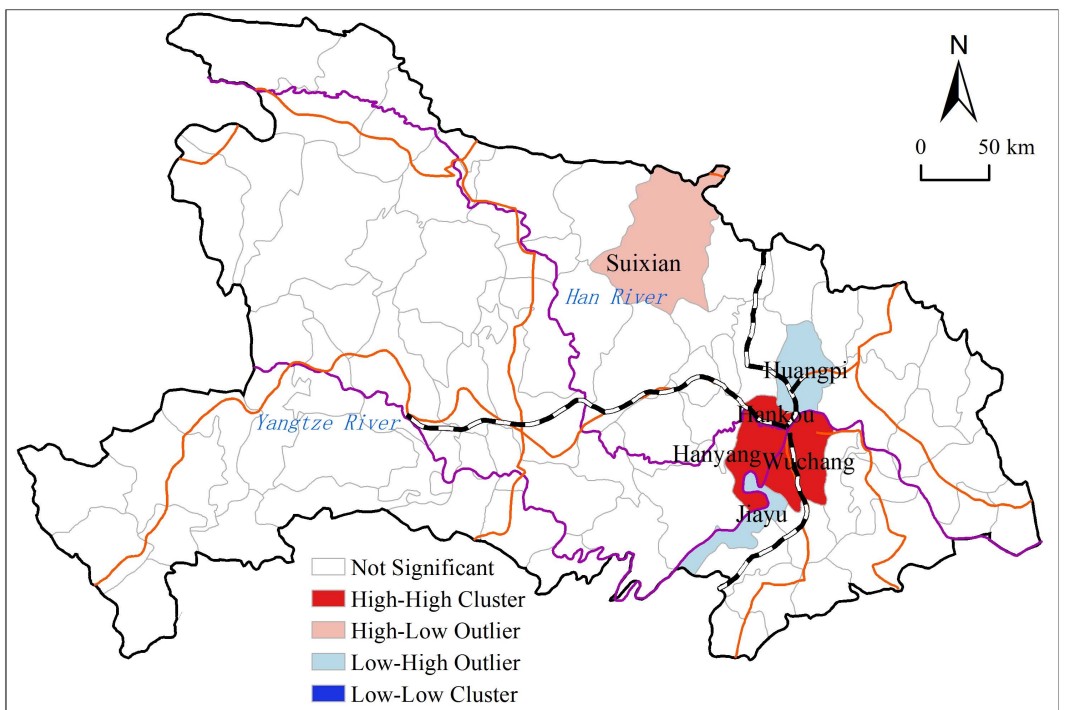

**Fig 6. Spatial clustering map of the smallpox epidemics in Hubei Province.** Note: The basemap came from United States Geological Survey (https://apps.nationalmap.gov/services/), the map boundary has not been changed. Cartographic software: ArcGIS.

**3.2.2. Spatial clustering.** Smallpox outbreaks in Hubei Province showed significant spatial clustering (Fig 6), with a Moran's I value of 0.2, p value < 0.01, and a Z value of 2.79. The distribution of smallpox epidemic in Hubei Province showed a clustered state. High-high clusters mainly distributed in Wuchang, Hankou, and Hanyang, which indicates these places were the hardest hit areas and high-risk areas of smallpox in Hubei Province. High-low clusters distributed only in Suixian, which located in the northern region of Hubei. Low-high clusters distributed in Jiayu and Huangpi, mainly in the areas around the three towns of Wuhan. There was no low-low clusters.

**3.2.3 Changes in the center of gravity.** The center of gravity of the smallpox epidemics in each decade from 1488 to 1949 was calculated (Fig 7), and the years 1488 to 1909, when there were fewer smallpox epidemics, were classified as one period. From 1488 to 1909, the epicenter of the epidemic was located in Jingshan. From 1910 to 1919, it moved northeast to Yingcheng. From 1920 to 1929, it moved southwest to Tianmen. In the following ten years, it continued to move southwest within Tianmen. From 1940 to 1949, the epicenter of the epidemic moved to the northwest. It fell into the territory of Jingmen. Overall, the center of gravity of smallpox in Hubei Province exhibited a small swing from east to west in the central region and gradually shifted westward, reflecting the gradual westward spread of the epidemic.

## 3.3. Regression analysis of factors influencing smallpox prevalence in Hubei Province

A quantitative analysis of factors influencing smallpox prevalence in Hubei Province was performed using geographically weighted regression models. Nine indicators were selected, namely, mean elevation ($x_1$), mean annual temperature ($x_2$), mean annual precipitation ($x_3$), river network density ($x_4$), number of years of floods ($x_5$), number of years of droughts ($x_6$), road network density ($x_7$), number of years of war ($x_8$), and population density ($x_9$). To eliminate the problem of multiple covariance, the value of the variance inflation factor (VIF) of each indicator was calculated by the OLS model, and the

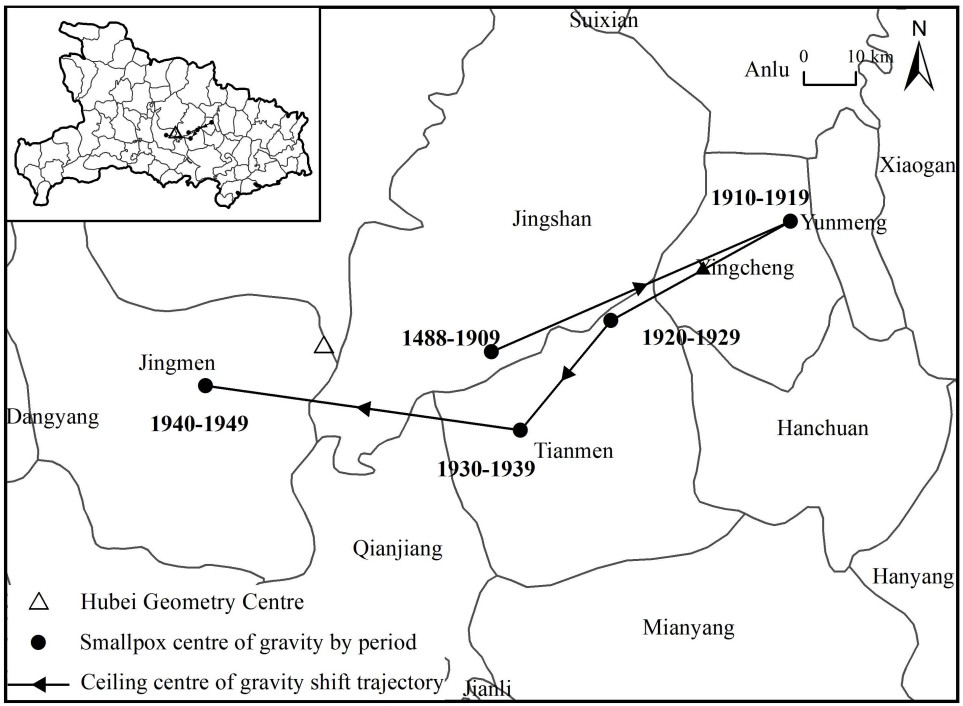

**Fig 7. Migration map of the smallpox epidemics center in Hubei Province.** Note: The basemap came from United States Geological Survey (https://apps.nationalmap.gov/services/), the map boundary has not been changed. Cartographic software: ArcGIS.

annual mean temperature ($x_2$) was removed. The geographically weighted regression tool in ArcGIS 10.7 was used to select the number of years of epidemic disease in each county as the dependent variable; the remaining eight indicators were used as independent variables. The FIXED kernel type was selected, and the range of the kernel was determined using the Akaike information criterion (AICc). The results show that, for goodness of fit, $R^2$=0.29, and the distribution of standardized residuals of the local regression model for each county is in the range [-1.91, 3.18], more than 96% of which are in the range [-2.58, 2.58]. These findings indicated that the standardized residuals of the geographically weighted regression model were random at the significance level of 4%, and only three counties fail the test of residuals; thus, the model fits well as a whole. The results showed that, from the perspective of positive and negative regression coefficients, the regression coefficients of four indicators, including river network density, road network density, number of years of war and population density, were all positive, indicating that they all had a positive effect on smallpox prevalence, and the regression coefficients of the indicators of the number of years of floods and the number of years of droughts were negative, indicating that they all had a negative effect on smallpox prevalence. In terms of the absolute size of the regression coefficients, the regression coefficients of the four indicators of river network density, road network density, the number of years of floods and the number of years of droughts were greater, indicating that they had a stronger effect on smallpox prevalence, and their regression coefficients were spatially visualized by the method of natural breakpoints (Fig 8) so that spatial variations in their influencing factors could be analyzed. Specifically, the regression coefficients of river network density (Fig 8a) and the number of years of war (Fig 8c) were positive and showed a spatial distribution of "low in the west and high in the east," indicating that river network and war had a promoting effect on smallpox prevalence and that this effect was more significant in eastern Hubei. River network and war factors both promoted smallpox epidemics and were more significant in eastern Hubei. The regression coefficient of the road network density (Fig 8b) index was positive, but

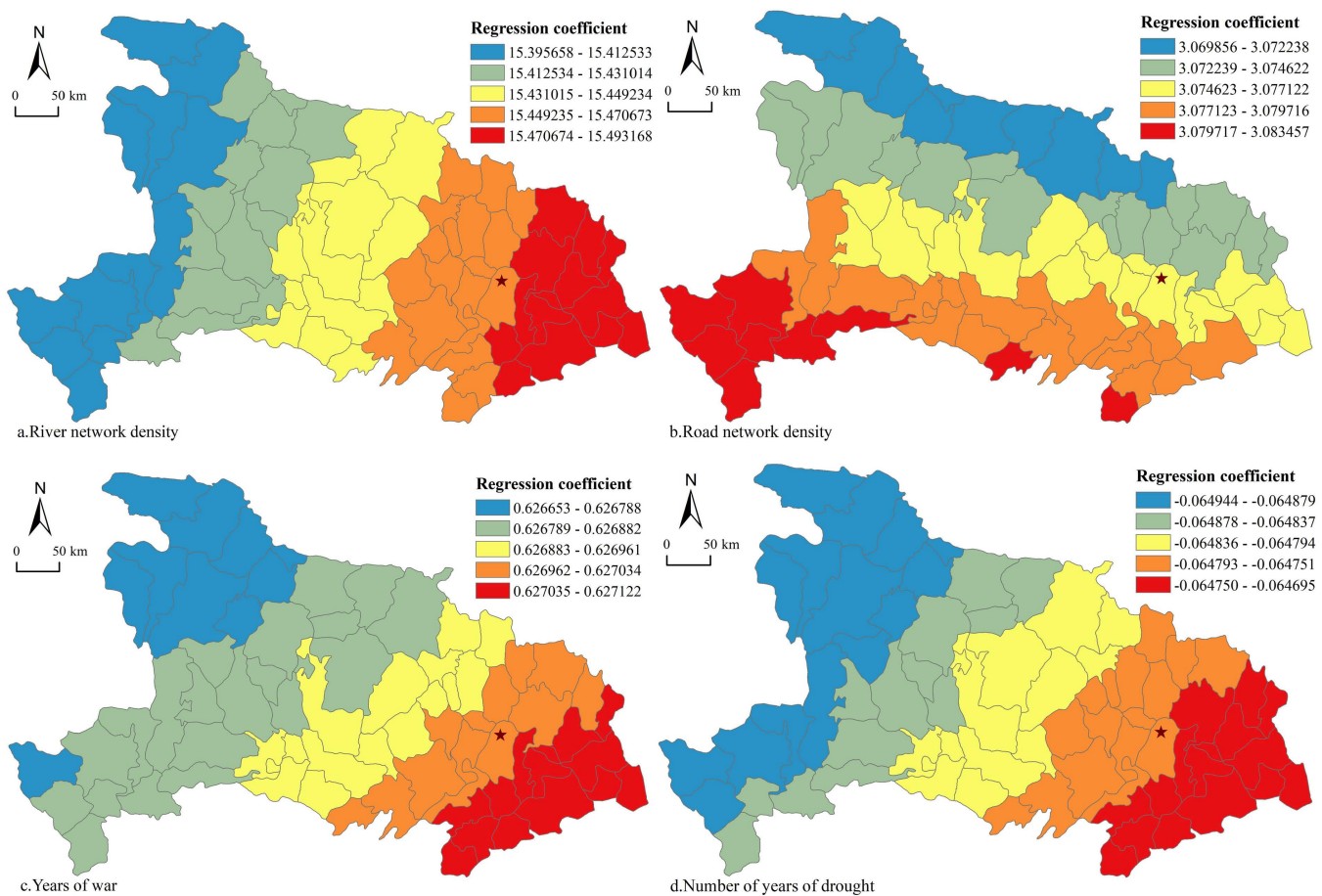

**Fig 8. Spatial variation in factors influencing smallpox epidemics in Hubei Province.** Note: (a) River network density. (b) Road network density. (c) Years of war. (d) Number of years of drought. The basemap came from United States Geological Survey (https://apps.nationalmap.gov/services/), the map boundary has not been changed. Cartographic software: ArcGIS.

it showed a spatial distribution characteristic of "high in the south and low in the north," indicating that the improvement of the road network in southern Hubei had a promoting effect on smallpox epidemics. In contrast, the regression coefficient for droughts years was negative and exhibited a spatial distribution characteristic of "high in the east and low in the west" (Fig 8d), indicating that droughts in western Hubei had a certain inhibitory effect on smallpox epidemics.

### 3.4. The formation mechanism of the spatial and temporal patterns of smallpox epidemics in Hubei Province

**3.4.1. Mechanisms shaping the temporal variation in smallpox epidemics in Hubei Province.** Temperature fluctuations, droughts and floods, and war outbreaks played dominant roles in the temporal characteristics of smallpox epidemics in Hubei Province. First, sudden changes in temperature and increased fluctuations can lead to an increase in epidemics [37]. Smallpox viruses are resistant to low temperature and dryness but not to humidity or heat; they are also transmitted through the respiratory tract, and most infections occur in spring and winter [38]. Temperature, relative humidity, and surface pressure have a positive impact on MPXV cases, dew/frost point, precipitation, and wind speed show a significant negative impact on MPXD cases [39]. Smallpox in Hubei Province was prevalent mainly in spring and summer, primarily because the temperature in spring is low, which is suitable for the survival of smallpox virus,

and fluctuates greatly, which reduces human immunity and increases the number of susceptible people. This situation in turn could lead to a smallpox epidemic, as stated in the "Declaration": "In 1901, in the spring of March (Wuchang), the weather was unusual, cold and warm were unpredictable, smallpox was prevalent, and many young children were infected. Even now that it is summer time, it is still not declining," [22]. Second, frequent droughts and floods in summer often indirectly trigger smallpox epidemics. Although the regression analysis results showed that the influence of droughts and floods years on smallpox epidemics was negatively correlated, historical data show that droughts and floods can easily induce smallpox. For example, the "Declaration" records, "Wuchang had been in a drought for a long time without rain, smallpox was prevalent and many people died," and the "Shishou County Chronicle" records, "In 1945, along various branches of the Ouchi River and the ancient long embankment in the north of the Yangtze River, a total of 13 places collapsed one after another, followed by epidemics of cholera, smallpox, and measles, killing more than 10,000 people," [22]. Third, the outbreak of large-scale wars would push smallpox epidemics to peaks. The regression analysis results showed that war years positively correlated with smallpox epidemics, with high regression coefficients. The Anti-Japanese War and the War of Liberation, which broke out during the 1930s and 1940s, were especially intense between 1938 and 1943, when the the four great battles of Wuhan, Suizao, Zaoyi, and Western Hubei occurred successively in Hubei Province. War destroyed normal life, increased population movement [40] and damaged urban structures; thus, it was extremely easy to contract epidemic diseases. For example, the "Gucheng County Chronicle" records, "In 1946, the armies of Wushan and Tianxin townships passed through the border. Many soldiers died in each army, and sick soldiers were everywhere along the way. As a result, the plague was prevalent, especially for malaria and typhoid fever. Nine out of the ten patients were infected, and the people were groaning in bed. It was very painful. The county data stated, according to the outpatient and inpatient statistics of the health center, 3,527 patients had smallpox disease, meningitis, typhoid dysentery, relapsing fever, or malaria throughout the year," [22]. The smallpox epidemic in Hubei Province showed an overall trend of "light at the beginning and heavy at the end" in terms of time changes, mainly concentrated in the Republic of China period, especially in the 1940s. Overall, it was mainly caused by the frequent wars and floods and droughts during this period.

### 3.4.2. Mechanisms shaping the spatial distribution of smallpox epidemics in Hubei Province.

Topography, population distribution and population movement played dominant roles in the spatial distribution pattern of smallpox prevalence in Hubei Province. First, the basic pattern of smallpox distribution in Hubei Province was determined by topography. The terrain of Hubei Province is sloped from northwest to southeast, surrounded by mountains in the east, west and north, and low in the central region, with a topographic structure of "seven mountains, one water and two fields." Additionally, the plains in the east and central parts of the province developed earlier, with dense populations and convenient transportation, which enabled more frequent smallpox epidemics. The western mountainous areas were developed later and were sparsely populated and relatively closed, hindering the prevalence and spread of smallpox. With the development of the western mountainous areas, smallpox epidemics also gradually increased, and the center of gravity of the epidemic tended to migrate to the west. Second, the population distribution determines the general pattern of smallpox distribution in Hubei. Smallpox patients are the only infectious source of smallpox, so smallpox outbreaks are closely related to population factors. The results of regression analyses showed that the effect of population density on smallpox epidemics was positively correlated, and the more densely populated an area was, the more easily smallpox spread. Population concentration also increased the number of infectious sources and transmission routes of smallpox epidemics, increasing the possibility of large-scale smallpox epidemics. The three towns of Wuhan, Suixian and Yichang were population centers in Hubei Province, with a large base of small-pox-susceptible people. Third, population movement reshaped the distribution pattern of smallpox strains in Hubei. Population movement led to the spread of the epidemic, with water and land transportation networks serving as the main media. The regression analysis results revealed that the influence of river network density and road network density on smallpox incidence were positively correlated, and the regression coefficient was high. Hubei Province is the "Throughway of Nine Provinces." The Yangtze

River is a highly utilized route for water transportation, and population exchanges are frequent. Smallpox spread along with the floating population at various waterway transportation hubs, causing the epidemic to spread within the province. Most of the areas prone to smallpox were located in important land and water transportation areas. For example, the Shengjing Times records, "In 1914, pox spread in Hankou and other ports along the river. Since the first month of this year, more than 2,000 people have died from the disease, and among others, the momentum is still not over yet," [22]. The "Xingshan County Chronicle" records, "In 1942, smallpox was prevalent, especially along the banks of the river in Gufu. Three people in a family died," [22]. In the 1930s and 1940s, the continuous outbreak of wars prompted a large number of military and political institutions and schools to migrate to Enshi in western Hubei Province, with a population of about 100,000, making Enshi the political, military and cultural center of Hubei Province. This was also an important reason for the migration of the center of gravity of smallpox from east to west.

## 4. Discussion

Epidemiological studies have shown that the incidence of smallpox before the universal vaccination of cowpox had a relatively regular cycle of increase; for example, in the seventeenth and eighteenth centuries in the U.S., there was an increase in the incidence every 2–3 years, and in 1900–1930 in India, there was an increase in the incidence every 5 years [41]. The present study revealed that there was a peak of smallpox epidemics every 6 or 12 years in Hubei Province during the Republic of China, which is consistent with this view. Existing studies have also focused on the relationship between smallpox incidence and demographic characteristics such as age and sex [42]. In this study, a total of 175 historical records of smallpox epidemics were collected, and 26 records on the age structure of smallpox-infected persons were found, 24 of which involved children infected with smallpox. It is evident that children and other people with lower immune levels were more susceptible to smallpox infection and that age characteristics can generally reflect the overall level of immunity and thus the size of the population susceptible to the smallpox virus. The scale of smallpox epidemics was related to factors such as settlement patterns and preventive measures [43]. Historically, people have used measures such as vaccination and isolation to prevent smallpox virus infection and to manage it. During this period, many medical books on pox diagnosis also emerged [44]. With the intervention of the government, special agencies and personnel were established to address smallpox, and vaccination was carried out in some cities. These actions suppressed the large-scale epidemic caused by smallpox to a certain extent and played an important role in the response to the public health crisis.

Although smallpox has been declared eradicated, various types of pox viruses still exist in the natural world, and these viruses are basically the same in terms of their antigenic nature and therefore have cross-tolerance. For example, monkeypox, which has been formally included in the management of China's category B statutory infectious diseases, belongs to the genus Orthopoxvirus, and its clinical manifestations are similar to those of smallpox, both of which are capable of "human-to-human" transmission [45]. Targeted preventive and control measures are therefore needed. First, in terms of time-varying characteristics, factors such as temperature fluctuations, droughts and floods, and war outbreaks play dominant roles; therefore, it is necessary to establish a comprehensive monitoring and early warning system for poxviruses such as smallpox and monkeypox and to strengthen monitoring and prevention in the winter and spring seasons, when temperature fluctuates considerably, and in the summer months, when floods are frequent, as well as during periods of war, to reduce the risk of smallpox epidemics. Second, from the perspective of spatial distribution patterns, factors such as terrain, population distribution, and population movement play a leading role. Areas with flat terrain, dense populations, and convenient transportation are prone to becoming epidemic centers. Therefore, densely populated large cities and transportation hubs should be used as epidemic monitoring sites. We should also focus on underdeveloped areas and border port cities with frequent population movements and strengthen the inspection and quarantine of transportation vehicles, people, livestock, goods, etc. Third, from the perspective of prevention and treatment drugs, human victory over smallpox is mainly attributed to the cowpox vaccine. The smallpox vaccine can induce cross-immunity for the monkeypox

virus. Vaccination with the smallpox vaccine can prevent monkeypox infection to a certain extent and reduce the severity of the disease. Therefore, a strategic stockpile of smallpox vaccines for emergency use should be produced and stored, and drugs to treat smallpox should be developed. Fourth, we must remain vigilant against bioterrorism, strengthen the management of smallpox virus, prevent smallpox virus from becoming a biological weapon, and conduct epidemic prevention exercises to respond to smallpox outbreaks.

Due to the lack of historical data, information on the early history of smallpox in Hubei Province is relatively scarce, and available information is incomplete. For example, there are records of smallpox epidemics in Yunmeng, Zhuxi, and Lishan counties, but information on the specific times of the outbreaks is unknown and therefore is not reflected in the spatial distribution map of the epidemics. Regarding the analysis of influencing factors and formation mechanisms, the consideration of influencing factors is not comprehensive enough, and the summary of formation mechanisms is still mainly a qualitative description. In the follow-up, the spatial and temporal patterns and formation mechanisms of smallpox epidemics in Hubei Province can be revealed in depth by further exploring smallpox and environmental historical data and introducing formation mechanism analysis methods such as machine learning.

## 5. Conclusions

(1) From the perspective of temporal changes, Hubei smallpox may be traced back to the Ming Dynasty. There are still few smallpox records from the Qing Dynasty. There were 9 smallpox years in the 268 years of the Qing Dynasty (1644–1911), for a frequency of 3.4%. In the Republic of China (1912–1949), there were 36 smallpox epidemics that occurred over 38 years, for a frequency of 94.74%. During the Republic of China period, smallpox showed a downward trend between 1912 and 1931, and an upward trend between 1931 and 1949. 1939 was the time when the smallpox epidemics began to suddenly increase.The smallpox epidemics in the Republic of China had two time scale fluctuation cycles: 8 years and 18 years. The main seasons for smallpox epidemics are spring and summer, followed by winter, and the least common is autumn.

(2) From the perspective of spatial distribution, smallpox epidemics in Hubei Province had a wide spatial scope and simultaneously showed the characteristics of spreading and proliferation; three towns of Wuhan and the Suixian and Yichang lines were the epidemic centers in the eastern, northern and western regions of Hubei, respectively. Smallpox epidemics in Hubei Province exhibited a significant spatial clustering pattern, with high and high clustering areas mainly distributed in Wuchang, Hankou and Hanyang; high and low clustering areas in Suixian in northern Hubei; and low and high clustering areas in Jiayu and Huangpi; these regions were distributed around the three towns of Wuhan and did not appear in the low and low clustering areas. The center of gravity of smallpox in Hubei Province exhibited a small swing from east to west in the central region and gradually shifted to the west, reflecting the gradual spread of the epidemic to the west and worsening of the disease.

(3) From the perspective of influencing factors and formation mechanisms, factors such as river networks, road networks, and wars all promoted smallpox epidemics. Among these factors, river network and war factors were significant in eastern Hubei, and road network factors were significant in southern Hubei. Therefore, droughts in western Hubei had a certain inhibitory effect on smallpox epidemics. Temperature fluctuations, droughts and floods, and outbreaks of war played leading roles in the temporal variation characteristics of smallpox epidemics in Hubei Province. Lower temperatures and increased fluctuations in spring led to increases in epidemics. Frequent droughts and floods in summer often indirectly trigger smallpox epidemics. The outbreak of large-scale wars would push smallpox epidemics to peaks. Topography, population distribution and population movement played dominant roles in the spatial distribution pattern of smallpox epidemics in Hubei Province, with topography revealing the basic pattern of smallpox distribution in Hubei Province, population distribution determining the overall pattern of smallpox distribution in Hubei Province, and population movement reshaping the pattern of smallpox distribution in Hubei Province.

As a recommendation, in the future, there should not be let up in the vigilance for the possibility of re-emergence of the smallpox virus, and continued attention should be given to the surveillance of smallpox and its analogous viruses in order to assess risk factors for smallpox transmission. Understanding the formation mechanism of smallpox epidemics from historical studies will help decision makers to be well prepared for future changes in the risk of smallpox and its similar viruses transmission.

## Supporting information

**S1 Table. Smallpox and Influencing factors data.**
(XLS)

## Author contributions

**Conceptualization:** Yuxin Zeng.

**Data curation:** Yuxin Zeng, Xihao Yan.

**Methodology:** Yuxin Zeng, Zhiyu Chen.

**Writing – original draft:** Yuxin Zeng, Zhiyu Chen, Xihao Yan.

**Writing – review & editing:** Shengsheng Gong, Tao Zhang.

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
