## [Decision Letter · Decision Letter 0]

9 Sep 2024

PONE-D-24-10506Geographical characteristics and formation mechanisms of smallpox epidemics in Hubei Province, China, 1488-1949PLOS ONE

Dear Dr. Zhang,

Thank you for submitting your manuscript to PLOS ONE. After careful consideration, we feel that it has merit but does not fully meet PLOS ONE’s publication criteria as it currently stands. Therefore, we invite you to submit a revised version of the manuscript that addresses the points raised during the review process.

Please provide a point by point response to the reviewer comments. 

We look forward to receiving your revised manuscript.

Kind regards,

Joanna Tindall, PhD

Staff Editor

PLOS ONE

Journal Requirements: When submitting your revision, we need you to address these additional requirements. 1. Please ensure that your manuscript meets PLOS ONE's style requirements, including those for file naming. The PLOS ONE style templates can be found at https://journals.plos.org/plosone/s/file?id=wjVg/PLOSOne_formatting_sample_main_body.pdf and https://journals.plos.org/plosone/s/file?id=ba62/PLOSOne_formatting_sample_title_authors_affiliations.pdf 2. Thank you for stating the following financial disclosure: "This work was supported by the National Natural Science Foundationof China (42371265�41801141),and the National Social Science Foundation�21VJXT015�". Please state what role the funders took in the study.  If the funders had no role, please state: ""The funders had no role in study design, data collection and analysis, decision to publish, or preparation of the manuscript."" If this statement is not correct you must amend it as needed. Please include this amended Role of Funder statement in your cover letter; we will change the online submission form on your behalf. 3. We note that your Data Availability Statement is currently as follows: All relevant data are within the manuscript and its Supporting Information files. Please confirm at this time whether or not your submission contains all raw data required to replicate the results of your study. Authors must share the “minimal data set” for their submission. PLOS defines the minimal data set to consist of the data required to replicate all study findings reported in the article, as well as related metadata and methods (https://journals.plos.org/plosone/s/data-availability#loc-minimal-data-set-definition). For example, authors should submit the following data: - The values behind the means, standard deviations and other measures reported;- The values used to build graphs;- The points extracted from images for analysis. Authors do not need to submit their entire data set if only a portion of the data was used in the reported study. If your submission does not contain these data, please either upload them as Supporting Information files or deposit them to a stable, public repository and provide us with the relevant URLs, DOIs, or accession numbers. For a list of recommended repositories, please see https://journals.plos.org/plosone/s/recommended-repositories. If there are ethical or legal restrictions on sharing a de-identified data set, please explain them in detail (e.g., data contain potentially sensitive information, data are owned by a third-party organization, etc.) and who has imposed them (e.g., an ethics committee). Please also provide contact information for a data access committee, ethics committee, or other institutional body to which data requests may be sent. If data are owned by a third party, please indicate how others may request data access.

Reviewers' comments:

Reviewer's Responses to Questions

**Comments to the Author**

1. Is the manuscript technically sound, and do the data support the conclusions?

Reviewer #1: Yes

Reviewer #2: Yes

2. Has the statistical analysis been performed appropriately and rigorously? 

Reviewer #1: Yes

Reviewer #2: Yes

3. Have the authors made all data underlying the findings in their manuscript fully available?

Reviewer #1: Yes

Reviewer #2: No

4. Is the manuscript presented in an intelligible fashion and written in standard English?

Reviewer #1: No

Reviewer #2: Yes

5. Review Comments to the Author

Reviewer #1: The provided research paper was an excellent attempt from the authors point of view small pox epidemic in Hubei provience,China but required some modification in the relevant literature with proper standarized reference style. In addition, the methods section must be modify with organometric and systematic proper mechanism and formation of epidemic pattern of small pox variation with chronological form

Reviewer #2: This article is written well with statistical data analysis. How ever, a few correction should be addressed by authors.

1. Please update data set up to date.

2. Clear research place of interest.

3. Why one M-K test is used without others. It would be better if authors use more model and compare them with validation. In this case Machine learning will be the best.

4. River networks, road networks, war, and other factors promoted smallpox epidemics. These are already established. Authors need to add more result-based evidence and data analysis to find out new outputs.

Specific comments:

Abstract: Conclusion needs to be edited as per main findings and its implications.

Introduction:

Please clear research gaps properly.

Authors need to write about virus also.

Methods:

Please add latitude and longitude.

Data files might be added in the supplementary files.a

Results:

Please add the probable reasons for epidemiological profile changes in Hubei Province, specifically, the scenario of 1940.

4-5 figures are suggested, please move to supply file or merge with one another.

6. PLOS authors have the option to publish the peer review history of their article (what does this mean? ). If published, this will include your full peer review and any attached files.

**Do you want your identity to be public for this peer review?** For information about this choice, including consent withdrawal, please see our Privacy Policy .

Reviewer #1: **Yes: ** Ashek Elahi Noor

Reviewer #2: No

---

## [Author Response · Author response to Decision Letter 1]

26 Oct 2024

We are very grateful to the two reviewers for their valuable comments. We have made detailed revisions based on their comments, which have been of great help in improving the quality of this article.

---

## [Decision Letter · Decision Letter 1]

3 Nov 2024

PONE-D-24-10506R1Geographical characteristics and formation mechanisms of smallpox epidemics in Hubei Province, China, 1488-1949PLOS ONE

Dear Dr. Zhang,

Thank you for submitting your manuscript to PLOS ONE. After careful consideration, we feel that it has merit but does not fully meet PLOS ONE’s publication criteria as it currently stands. Therefore, we invite you to submit a revised version of the manuscript that addresses the points raised during the review process.

Reviewer 1

Review of Manuscript PONE-D-24-10506: "Geographical characteristics and formation mechanisms of smallpox epidemics in Hubei Province, China, 1488-1949"

The manuscript provides a comprehensive analysis of the spatial and temporal distribution of smallpox epidemics in Hubei Province, China, from 1488 to 1949. The study employs rigorous methodologies, including the M-K test, wavelet analysis, spatial autocorrelation, and geographically weighted regression models, to identify key environmental and social factors that influenced epidemic patterns. The historical insights and advanced statistical tools applied make this a valuable contribution to understanding epidemic patterns in historical contexts.

Suggested Revisions:

Clarity in Methodology: While the methodology is sound, further clarification on certain models, such as the specifics of the wavelet analysis and M-K test implementation, would enhance reproducibility for readers unfamiliar with these techniques.

Interpretation of Results: The section on the spatial shift of the epidemic center (east to west) could benefit from a deeper analysis of underlying socio-political or environmental factors to better contextualize this finding.

Discussion on Prevention and Control Implications: The conclusion briefly mentions preventive measures against smallpox and similar viruses. Expanding on this, particularly with lessons learned from historical epidemics, could provide valuable insights for modern epidemic management.

Conclusion: This manuscript is a valuable addition to historical epidemiological studies and enhances understanding of smallpox spread dynamics in historical China. With minor revisions to methodology and discussion, this paper would be a strong candidate for publication.

Reviewer 2

Abstract

Please mention data collection time and source.

Please input your significant results in abstract.

Introduction

Please add references: Smallpox originally originated in ancient Egypt or India 3,000 years ago. (https://doi.org/10.1016/j.imj.2023.11.001)

Please add research gap regarding the issues.

Add more epidemiology from . https://doi.org/10.22207/JPAM.16.SPL1.16;
https://doi.org/10.3390/ijerph192315638

Methodology:

Justification is required for using Mann-Kendall (M-K) tests, wavelet analysis, and geographically weighted regression (GWR) and parameter selection for these methods, such as the bandwidth used in GWR and justification for specific thresholds in the M-K test.

Manuscript used multiple data sources however there is lack of clarity on data integration techniques.

Please write on data cleaning and processing.

Result

Please add why some regions or time periods showed higher or lower smallpox incidence.

Moran’s I and local Moran’s I findings showed statistically significant. Pleas add more practical implications.

Reviewer 3

Accept

We look forward to receiving your revised manuscript.

Kind regards,

Sara Hemati

Academic Editor

PLOS ONE

Journal Requirements:

**Comments to the Author**

1. If the authors have adequately addressed your comments raised in a previous round of review and you feel that this manuscript is now acceptable for publication, you may indicate that here to bypass the “Comments to the Author” section, enter your conflict of interest statement in the “Confidential to Editor” section, and submit your "Accept" recommendation.

Reviewer #1: All comments have been addressed

Reviewer #2: (No Response)

Reviewer #3: (No Response)

2. Is the manuscript technically sound, and do the data support the conclusions?

Reviewer #1: Yes

Reviewer #2: No

Reviewer #3: (No Response)

3. Has the statistical analysis been performed appropriately and rigorously? 

Reviewer #1: Yes

Reviewer #2: (No Response)

Reviewer #3: (No Response)

4. Have the authors made all data underlying the findings in their manuscript fully available?

Reviewer #1: Yes

Reviewer #2: Yes

Reviewer #3: (No Response)

5. Is the manuscript presented in an intelligible fashion and written in standard English?

Reviewer #1: Yes

Reviewer #2: Yes

Reviewer #3: (No Response)

6. Review Comments to the Author

Reviewer #1: Review of Manuscript PONE-D-24-10506: "Geographical characteristics and formation mechanisms of smallpox epidemics in Hubei Province, China, 1488-1949"

The manuscript provides a comprehensive analysis of the spatial and temporal distribution of smallpox epidemics in Hubei Province, China, from 1488 to 1949. The study employs rigorous methodologies, including the M-K test, wavelet analysis, spatial autocorrelation, and geographically weighted regression models, to identify key environmental and social factors that influenced epidemic patterns. The historical insights and advanced statistical tools applied make this a valuable contribution to understanding epidemic patterns in historical contexts.

Suggested Revisions:

Clarity in Methodology: While the methodology is sound, further clarification on certain models, such as the specifics of the wavelet analysis and M-K test implementation, would enhance reproducibility for readers unfamiliar with these techniques.

Interpretation of Results: The section on the spatial shift of the epidemic center (east to west) could benefit from a deeper analysis of underlying socio-political or environmental factors to better contextualize this finding.

Discussion on Prevention and Control Implications: The conclusion briefly mentions preventive measures against smallpox and similar viruses. Expanding on this, particularly with lessons learned from historical epidemics, could provide valuable insights for modern epidemic management.

Conclusion: This manuscript is a valuable addition to historical epidemiological studies and enhances understanding of smallpox spread dynamics in historical China. With minor revisions to methodology and discussion, this paper would be a strong candidate for publication.

Reviewer #2: Abstract

Please mention data collection time and source.

Please input your significant results in abstract.

Introduction

Please add references: Smallpox originally originated in ancient Egypt or India 3,000 years ago. (https://doi.org/10.1016/j.imj.2023.11.001)

Please add research gap regarding the issues.

Add more epidemiology from . https://doi.org/10.22207/JPAM.16.SPL1.16;
https://doi.org/10.3390/ijerph192315638

Methodology:

Justification is required for using Mann-Kendall (M-K) tests, wavelet analysis, and geographically weighted regression (GWR) and parameter selection for these methods, such as the bandwidth used in GWR and justification for specific thresholds in the M-K test.

Manuscript used multiple data sources however there is lack of clarity on data integration techniques.

Please write on data cleaning and processing.

Result

Please add why some regions or time periods showed higher or lower smallpox incidence.

Moran’s I and local Moran’s I findings showed statistically significant. Pleas add more practical implications.

Reviewer #3: The manuscript on the Geographical Characteristics and Formation Mechanisms of Smallpox Epidemics in Hubei Province, China, 1488-1949 presents a well-researched and insightful analysis of the historical epidemiology of smallpox in a crucial region of China. By examining a broad historical timeline, the authors successfully illustrate the spatial and environmental factors that influenced epidemic patterns, providing a nuanced understanding of how geographical and social factors affected the spread and persistence of smallpox over several centuries. This research not only contributes valuable data to historical epidemiology but also enhances our comprehension of the interplay between geography and disease dynamics in pre-modern China. Overall, the study is a significant contribution to the field, offering both depth in historical context and clarity in spatial analysis.

7. PLOS authors have the option to publish the peer review history of their article (what does this mean? ). If published, this will include your full peer review and any attached files.

**Do you want your identity to be public for this peer review?** For information about this choice, including consent withdrawal, please see our Privacy Policy .

Reviewer #1: **Yes: ** Ashek Elahi Noor

Reviewer #2: **Yes: ** MD AMINUL ISLAM

Reviewer #3: No

---

## [Author Response · Author response to Decision Letter 2]

18 Dec 2024

We are very grateful for the valuable comments from the reviewers. We have made corresponding modifications and improvements based on them.

---

## [Editor Report · Decision Letter 2]

22 Dec 2024

Geographical characteristics and formation mechanisms of smallpox epidemics in Hubei Province, China, 1488-1949

PONE-D-24-10506R2

Dear Dr. Zhang,

We’re pleased to inform you that your manuscript has been judged scientifically suitable for publication and will be formally accepted for publication once it meets all outstanding technical requirements.

Kind regards,

Sara Hemati

Academic Editor

PLOS ONE
---

## [Editor Report · Acceptance letter]

PONE-D-24-10506R2

PLOS ONE

Dear Dr. Zhang,

I'm pleased to inform you that your manuscript has been deemed suitable for publication in PLOS ONE. Congratulations! Your manuscript is now being handed over to our production team.

Kind regards,

on behalf of

Dr. Sara Hemati

Academic Editor

PLOS ONE